# Investigation of the Optimal Immunization Dose and Protective Efficacy of an Attenuated and Marker *M. bovis*–Bovine Herpesvirus Type 1 Combined Vaccine in Rabbits

**DOI:** 10.3390/ani14050748

**Published:** 2024-02-28

**Authors:** Sen Zhang, Guoxing Liu, Wenying Wu, Li Yang, Ihsanullah Shirani, Aizhen Guo, Yingyu Chen

**Affiliations:** 1National Key Laboratory of Agricultural Microbiology, Hubei Hongshan Laboratory, College of Veterinary Medicine, Huazhong Agricultural University, Wuhan 430070, Chinai.shirani786@webmail.hzau.edu.cn (I.S.); 2Hubei International Scientific and Technological Cooperation Base of Veterinary Epidemiology, The Cooperative Innovation Center for Sustainable Pig Production, Wuhan 430070, China; 3Key Laboratory of Development of Veterinary Diagnostic Products, Ministry of Agriculture and Rural Affair, Wuhan 430070, China; 4Wuhan Keqian Biology Co., Ltd., Wuhan 430200, China

**Keywords:** BRD, *M. bovis*, BoHV-1, combined vaccine, immunization dose

## Abstract

**Simple Summary:**

Bovine respiratory disease (BRD) is a globally prevalent multifactorial infection primarily caused by viral, bacterial, and *Mycoplasma* coinfections. Vaccines exist for individuals or several pathogens, but there are currently no live and marker combined vaccines commercially available. Based on our previous study, we developed an attenuated and marker *M. bovis*–BoHV-1 combined vaccine for the first time and determined the optimal antigen ratio of the combined vaccine. In this work, we exhibited different vaccine doses to further determine the optimal immunization dose using a rabbit model. The findings indicated that the 2:2 immunization dose group showed the best performance in humoral and cellular immune responses. Also, it had a nearly standard lung tissue structure and was able to provide the best protective efficacy in facing the challenge with both pathogens. The results of this study determined the optimal immune dose for the combined vaccine. Combined with our previous research, these findings lay the foundation for further bovine experiments and actual clinical applications of the attenuated and marker *M. bovis*–BoHV-1 combined vaccine.

**Abstract:**

Bovine respiratory disease (BRD) is one of the most common diseases in the cattle industry; it is a globally prevalent multifactorial infection primarily caused by viral and bacterial coinfections. In China, *Mycoplasma bovis* (*M. bovis*) and bovine herpesvirus type 1 (BoHV-1) are the most notable pathogens associated with BRD. Our previous study attempted to combine the two vaccines and conducted a preliminary investigation of their optimal antigenic ratios. Based on this premise, the research extended its investigation by administering varying vaccine doses in a rabbit model to identify the most effective immunization dosage. After immunization, all rabbits in other immunization dose groups had a normal rectal temperature without obvious clinical symptoms. Furthermore, assays performed on the samples collected from immunized rabbits indicated that there were increased humoral and cellular immunological reactions. Moreover, the histological analysis of the lungs showed that immunized rabbits had more intact lung tissue than their unimmunized counterparts after the challenge. Additionally, there appears to be a positive correlation between the protective efficacy and the immunization dose. In conclusion, the different immunization doses of the attenuated and marker *M. bovis* HB150 and BoHV-1 gG-/tk- combined vaccine were clinically safe in rabbits; the mix of 2.0 × 10^8^ CFU of *M. bovis* HB150 and 2.0 × 10^6^ TCID_50_ BoHV-1 gG-/tk- strain was most promising due to its highest humoral and cellular immune responses and a more complete morphology of the lung tissue compared with others. These findings determined the optimal immunization dose of the attenuated and marker *M. bovis* HB150 and BoHV-1 gG-/tk- combined vaccine, laying a foundation for its clinical application.

## 1. Introduction

Bovine respiratory disease (BRD) is a widespread cause of morbidity, mortality, and economic loss in beef and dairy cattle farming [1]. BRD may also be responsible for up to 70% of morbidity and mortality in U.S. feedlot cattle [2,3], and BRD is recognized as the primary disease affecting cattle in over 90% of feedlots across the United States, with an estimated annual cost ranging from 1 to 3 billion USD [4,5]. In US feedlots, it costs $23.6 to treat each cattle with respiratory diseases, and a recent study estimated the cost per BRD-affected calf to be as high as $42.15 [6]. In Australia, the recorded rates of illness and death for BRD were 18% and 2.1%, respectively, with an average financial loss of $1076.17 per death [7]. BRD is a multifactorial disease, and its etiology involves bacterial and viral coinfections with high re-infection rates [8]; also, the susceptibility of cattle to this disease can be influenced by a complex interaction between stress, management practices, physiological conditions, and the host immune response [1,9,10]. Moreover, immunosuppression caused by viral infections is also significantly associated with the risk of severe secondary bacterial infections.

Common pathogens associated with BRD include bovine herpesvirus type 1 (BoHV-1) and bovine viral diarrhea virus, *Mycoplasma bovis* (*M. bovis*), *Mannheimia haemolytica*, and *Pasteurella multocida* [11]. In China, *M. bovis* and BoHV-1 are the pathogens most frequently associated with BRD [12]. *M. bovis* was initially identified in 2008 among recently imported beef cattle suffering from pneumonia, and its prevalence has become widespread [13]. BoHV-1 was initially detected in 1981 in an imported cow and quickly disseminated nationwide. Approximately 40% of cattle in China were found to be infected with BoHV-1 [14]. In Inner Mongolia and Henan Province, the nucleic acid positivity rate of BoHV-1 could reach 7% to 15% and 24.83%, respectively [9,14]. Vaccines exist for individuals or several pathogens; Bimeda Biochemicals and Boehringer Ingelheim Animal Health have developed live vaccines to target specific pathogens of BRD, such as *M. bovis* and BoHV-1. Nonetheless, there are currently no live and marker combined vaccines commercially available.

In our previous study, we developed an attenuated and marker *M. bovis*–BoHV-1 combined vaccine for the first time, and the results showed that the combined vaccine provided sufficient and effective protection for rabbits when facing the challenge with both pathogens. Moreover, the protective effect was best at a 1:1 ratio of the two antigens in all vaccinated groups when challenged with *M. bovis* HB0801 or BoHV-1 HB06 [12,15,16]. However, its protective efficacy has not been quantitatively evaluated. To enhance the immunoprotective rate of the attenuated and marker *M. bovis*–BoHV-1 combined vaccine, different vaccine doses were developed to further determine the optimal immunization dose using a rabbit model in this study and evaluated the vaccine’s protective efficacy by quantitative scores. Our study provides a solid foundation for clinically applying the attenuated and marker *M. bovis*–BoHV-1 combined vaccine.

## 2. Materials and Methods

### 2.1. Cells and Viruses

The wild-type BoHV-1 strain HB06 (GenBank accession number: AJ004801.1), along with the BoHV-1 gG-/tk- strain and *M. bovis* strains HB0801 (GenBank accession number: CP002058.1) and HB150, was preserved at the State Key Laboratory of Agricultural Microbiology. Madin–Darby bovine kidney cells (MDBK) were obtained from the China Institute of Veterinary Drug Control.

### 2.2. Culture of M. bovis and BoHV-1

*M. bovis* and BoHV-1 were cultured as previously described. Briefly, *M. bovis* HB0801 and HB150 strains were cultured in a complete PPLO medium at 37 °C in a 5% CO_2_ incubator for 40–48 h. BoHV-1 HB06 and gG-/tk- strains were cultured in Dulbecco’s modified Eagle’s medium (DMEM) supplemented with 10% fetal bovine serum (Inner Mongolia Opcel Biotechnology Co., Ltd.) using MDBK cells at 37 °C in a 5% CO_2_ incubator.

The *M. bovis* HB150 and BoHV-1 gG-/tk- initial doses were 1.0 × 10^8^ CFU and 1.0 × 10^6^ TCID_50_, respectively. The ratios of 1:1, 2:2, and 3:3 were prepared as follows: 1.0 × 10^8^ CFU *M. bovis* HB150 with 1.0 × 10^6^ TCID_50_ BoHV-1 gG-/tk-, 2.0 × 10^8^ CFU *M. bovis* HB150 with 2.0 × 10^6^ TCID_50_ BoHV-1 gG-/tk-, and 3.0 × 10^8^ CFU *M. bovis* HB150 with 3.0 × 10^6^ TCID_50_ BoHV-1 gG-/tk-.

### 2.3. Animal Experiments

A group of 27 Japanese white rabbits, with an average weight of 1.5 kg each, were found to be seronegative for *M. bovis*, BoHV-1, *Pasteurella*, and *Manniella hemolysticus* and were divided into 9 groups. All rabbits were housed in isolation to prevent cross-infection. According to the antigen dose, groups 1 and 2 were immunized with 1.0 × 10^8^ CFU *M. bovis* HB150 combined with 1.0 × 10^6^ TCID_50_ BoHV-1 gG-/tk- strain (presented as 1:1); groups 3 and 4 were immunized with 2.0 × 10^8^ CFU *M. bovis* HB150 mixed with 2.0 × 10^6^ TCID_50_ BoHV-1 gG-/tk- strain (presented as 2:2); groups 5 and 6 were immunized with 3.0 × 10^8^ CFU *M. bovis* HB150 mixed with 3.0 × 10^6^ TCID_50_ BoHV-1 gG-/tk- strain (presented as 3:3), respectively; groups 7 and 8 were inoculated with PPLO complete medium and DMEM, and group 9 served as a control. The design of the study is outlined in detail in Table 1. All experimental groups were then challenged with 1.0 × 10^9^ CFU *M. bovis* HB0801 strain or 4.0 × 10^7^ TCID_50_ BoHV-1 HB06 strain 28 days after immunization. Nasal swabs were collected daily for 14 days after immunization and challenge, and blood samples were collected weekly until the experiment’s end. Samples were stored at −80 °C. Animals were euthanized at the end of the investigation, and lung tissue samples were collected for further experiments.

### 2.4. Clinical Evaluation and Sample Collection

Rectal temperature and clinical observation were recorded for 14 days following immunization and the challenge. A thermometer was inserted approximately 4 cm into the rectum until the temperature stopped changing. The rectal temperature of rabbits was measured in the morning and afternoon, and the average value was recorded.

After immunization and challenge, nasal swabs were obtained daily for 14 days. The samples were thoroughly mixed by vortexing in tubes with 1 mL of sterile PBS, then filtered through a 0.45 µm filter, and preserved at −20 °C for subsequent PCR/RT-PCR analysis and *M. bovis* quantification. Blood samples were collected weekly for antibody and cytokine detection until the end of the experiment. Lung samples were collected 28 days after the challenge. The tissue samples were preserved in a 4% paraformaldehyde solution for 48 h and subsequently encased in paraffin. The sections were subjected to hematoxylin–eosin staining and later underwent histopathological evaluation.

### 2.5. Virus and Bacteria Shedding

The total nucleic acids of nasal swabs were extracted and amplified by PCR/RT-PCR to detect the shedding of *M. bovis* HB150 using the *uvrC* gene and BoHV-1 gG-/tk- using glycoprotein G *gG* and thymidine kinase *tk* genes, as previously described under the following reaction conditions: *M. bovis* uvrC gene PCR reaction conditions: 95 °C for 3 min, 95 °C for 15 s, 55 °C for 15 s, 72 °C for 30 s, 35 cycles; 72 °C for 5 min; BoHV-1 gG PCR reaction conditions: 95 °C for 3 min, 95 °C for 15 s, 60.5 °C for 15 s, 72 °C for 40 s, 35 cycles, 72 °C for 5 min; BoHV-1 tk PCR reaction conditions: 95 °C for 3 min; 95 °C for 15 s, 60.5 °C for 15 s, 72 °C for 30 s, 35 cycles, 72 °C for 5 min. The primers used in this study are listed in Table 2.

To measure the shedding of *M. bovis* HB0801 post-challenge, the nasal swabs subjected to treatment were progressively diluted by a factor of 10. Subsequently, 100 μL from each dilution was added to a PPLO liquid medium and incubated at 37 °C in a 5% CO_2_ incubator for 3–5 days or until no further change in the medium color occurred. The highest dilution showing a color change was 1 CCU/mL. The counting of *M. bovis* HB0801 shedding was then determined based on the dilution factor. BoHV-1 HB06 shedding after the challenge was detected from the DNA extracted from nasal swabs used for RT-PCR using the envelope glycoprotein B *gB* gene. The program for BoHV-1 *gB* gene was as follows: 95 °C for 30 s, 95 °C for 10 s, 60 °C for 20 s, 40 cycles; 95 °C for 15 s, 60 °C for 20 s, 95 °C for 15 s.

### 2.6. Serum Antibody of M. bovis

Serum antibodies against *M. bovis* were detected using competitive ELISA, following established protocols. Briefly, the test serum, positive control, and negative control sera, all diluted fourfold, were added to plates coated with *M. bovis* p579 protein and HRP-labeled monoclonal antibodies. The plates were then incubated at 37 °C for 60 min. After washing, 100 µL of chromogenic substrate solution was added and incubated at room temperature, shielded from light, for 10 min. The OD_450 nm_ was promptly measured upon halting the reaction. The blocking rate (P.I. value) was calculated as follows:

Blocking rate = (1 − S/N) × 100%, S = sample OD_450 nm_; N = mean OD_450 nm_ of negative control serum. Conditions for the establishment of the test: 0.65 < OD_450 nm_ negative control < 2.0, PI_positive control_ > 60%. PI_sample_ ≥ 41% means positive; PI_sample_ < 41% means negative.

### 2.7. Neutralization Assay

Inactivated serum (56 °C for 30 min) was serially diluted in a 96-well cell culture plate and then incubated with 100 TCID_50_ BoHV-1 HB06 virus at 37 °C in a 5% CO_2_ incubator for 1 h. The serum–virus mixture was then transferred to a 96-well cell culture plate containing MDBK cells and cultured in a 5% incubator at 37 °C for three days. Neutralizing antibody titers, the highest serum dilutions that inhibit BoHV-1 infection, were calculated using the Reed–Muench method.

### 2.8. Detection of Cytokines and ELISA Antibodies

Commercial ELISA kits (Jiangsu Meimian Industrial Co., Ltd., Yancheng, China) were used to detect changes in serum cytokine and antibody levels, including IL-1β, IL-6, TNF-α, IFN-γ, sIgA, and IgG.

### 2.9. Calculation of Protective Efficacy

The immunoprotective rate of the attenuated and marker *M. bovis*–BoHV-1 combined vaccine after challenging with *M. bovis* HB0801 and BoHV-1 HB06 was calculated using the following equation:$$PE=\frac{\left(S_{PC}-S_{NC}\right)-(S_{VAC}-S_{NC})}{\left(S_{PC}-S_{NC}\right)}\times 100\%$$

*S_PC_*, *S_VAC_*, and *S_NC_* represent the mean of the sum of clinical symptom scores and lung pathological changes in the rabbits in the non-immune challenge group, immune challenge group, and control group, respectively.

### 2.10. Statistical Analysis

Student’s *t*-test and one-way ANOVA were used to detect significant differences between groups, where *p* < 0.05 (*), *p* < 0.01 (**), *p* < 0.001 (***), or *p* < 0.0001 (****) were considered to be statistical differences. * represents there exist a difference, but the difference is not significant; **, *** and **** represent the significant degree of the differences, respectively. Error bars indicate the standard error of the mean.

## 3. Results

### 3.1. Clinical Signs

Most rabbits in the immunized groups had normal rectal temperature without obvious respiratory signs during the entire observation period; only some underwent slight rectal temperature elevation one or two days after immunization and returned to normal (Figure 1A). In the unimmunized but challenged groups, the rectal temperatures of the rabbits surpassed 39.5 °C on days 1–3 and 2–4 following the challenge with BoHV-1 HB06 and *M. bovis* HB0801, respectively (Figure 1B,C). All animals in these two groups showed clinical signs such as coughing, nasal secretion, and ocular secretions. These results confirmed that rabbits were successfully challenged with *M. bovis* HB0801 and BoHV-1 HB06 and that the developed vaccine was safe.

### 3.2. Detection of M. bovis and BoHV-1 Shedding

#### 3.2.1. *M. bovis* HB150 and BoHV-1 gG-/tk- Shedding after Immunization

Assays performed on the samples collected from groups immunized with an immunization dose 1:1 showed that rabbits stopped shedding *M. bovis* HB150 on the 5th day after immunization, and the 2:2 and 3:3 groups stopped shedding on the 6th and 7th days, respectively. In addition, the analysis showed that in the 1:1 group, over 50% of the animals exhibited continuous shedding on day 3 post-immunization. The condition lasted one more day in the 2:2 and 3:3 immunization groups (Table 3).

When the collected nasal swabs were collected and analyzed, we found that no shedding of BoHV-1 gG-/tk- was detected in the 2:2 group on day 8 post-immunization, and the 1:1 and 3:3 groups stopped shedding 9 days post-immunization. The control group consistently showed no signs of viral shedding during observation. On day 5 post-immunization, more than 50% of animals in all immunized groups continued to shed (Table 3).

Therefore, among all mixed immunization groups, the 3:3 mixed immunization group exhibited the highest and most sustained levels of shedding for both pathogens.

#### 3.2.2. *M. bovis* HB0801 and BoHV-1 HB06 Shedding after Challenge

Following the challenge with *M. bovis* HB0801, nasal swab samples were collected and analyzed everyday, and the conclusion was that only a small quantity of *M. bovis* HB0801 shedding was observed in the 1:1 group on the initial day, and no shedding was detected on the second day post-challenge. During the initial 1–2 days, all mixed-vaccine immunization groups displayed a decrease in *M. bovis* HB0801 shedding compared to the non-immune-challenged group. Despite this, no statistically significant differences were observed between the vaccinated and non-immune-challenged groups during this period (Figure 2A).

BoHV-1 HB06 shedding after the challenge was measured using RT-PCR. When we processed and analyzed the nasal swabs samples collected daily, we found that high titers of BoHV-1 HB06 were detected in the non-immune-challenged group from day 3 to day 6 and peaked on day 4 after the challenge. In contrast, in different immunization dose groups, all immunized groups exhibited lower BoHV-1 HB06 shedding compared to the nonimmune-challenged group on days 3–5 after the challenge (*p* < 0.05). Importantly, no significant difference was found between immunized groups during the entire observation period (Figure 2B). Ultimately, the 1:1 group exhibited the lowest shedding compared to the other mixed immunization groups following the challenge with either *M. bovis* HB0801 or BoHV-1 HB06.

### 3.3. Antibody Response

#### 3.3.1. *M. bovis* Serum Antibody Levels

Competitive ELISA was used to measure the level of *M. bovis* serum antibodies. The antibody level of the 2:2 and 3:3 groups was significantly higher than that of the control from days 7 to 21 after immunization (*p* < 0.05), and the blocking rate reached 48.6% and 51.9% at day 14 after immunization, respectively. From day 7 to 14 post-challenge, the 2:2 and 1:1 groups displayed substantially higher antibody titer compared to the non-immunized but challenged groups, respectively (*p* < 0.05) (Figure 3A); at 14 days post-challenge, the 1:1 group had the highest blocking rate during the entire observation period, which could reach 77.9%.

#### 3.3.2. BoHV-1-Neutralizing Antibody Response

BoHV-1-neutralizing antibodies in rabbits were quantified using a serum neutralization assay. All immunized groups found low levels of neutralization titer until 7 days post-challenge; at this time, the 1:1 and 2:2 groups induced significantly higher neutralization titer than the non-immunized but challenged group (*p* < 0.05). At 14 days after the challenge, the 1:1 group caused the highest levels of neutralizing antibody titer in all experimental groups during the observation period, which reached 1:22. Rabbits in the control group did not produce any neutralizing antibodies throughout the experimental period (Figure 3B). In summary, the 1:1 mixed-immunization group can induce the highest and most sustained *M. bovis* and BoHV-1 antibody titers among all immunization groups.

### 3.4. M. bovis–BoHV-1 Combined Vaccine Induces Cellular Immunity in Rabbits

After vaccination, the TNF-α and IFN-γ levels were significantly higher in all immunized groups, and levels of IL-1β and IL-6 slightly increased. Still, the levels of these cytokines in the immunized groups were considerably higher than those in the control group (*p* < 0.01). Notably, the 2:2 group showed levels as high as 250 μg/mL and 200 μg/mL of TNF-α and IFN-γ from 7 to 14 days post-immunization, respectively, inducing the most robust cellular immune response of all groups. Thus, after immunization, the *M. bovis*–BoHV-1 combined vaccine represented by the 2:2 group induced a mixed Th1/Th2 response biased toward Th1 in rabbits (Figure 4A–D).

### 3.5. sIgA and IgG Titers in Rabbits

To evaluate the vaccine-induced systemic immune response, serum IgA and IgG antibody levels were measured. The vaccine can produce sIgA antibodies after immunization, with the highest antibody levels induced in the 2:2 group, indicating that the vaccine can cause a mucosal immune response. The sIgA antibody levels in all vaccinated groups were significantly higher than those of the control during the whole observation period (*p* < 0.0001) (Figure 5A).

IgG plays a key role in the secondary immune response. Here, vaccinated rabbits rapidly produced high levels of IgG antibodies after the challenge. IgG antibody levels dramatically increased in the immunized groups challenged by *M. bovis* HB0801, with the most pronounced elevation to 220 μg/mL in the 2:2 group at 21 days post-challenge. During the whole observation period, the IgG levels in all vaccinated groups were significantly higher than those in the control group (*p* < 0.01) (Figure 5B,C). Following exposure to BoHV-1 HB06, the immunized groups exhibited elevated levels of IgG antibodies. Notably, the IgG levels induced in all vaccinated groups were significantly higher than the control group during the experimental period (*p* < 0.01). Overall, the 2:2 group induced the highest levels of sIgA and IgG antibodies after immunization and challenge, respectively.

### 3.6. Evaluation of Micro Pathological Injury

After *M. bovis* HB0801 challenge, the alveolar structure was relatively complete, with no apparent pathological alterations in the alveolar wall and alveolar cavity. Only a small amount of plasmacytic exudation with inflammatory cells was found in the alveolar cavity in immunized groups, and there were varying degrees of alveolar wall hyperplasia but little disruption of the structure; especially in the 3:3 group, after observing the lung structure in this experimental group, we found that the alveolar structure was slightly damaged. In contrast, after analysis of the lung structure of unvaccinated rabbits challenged with *M. bovis,* HB0801 presented the most severe lesions and loss of normal lung tissue structure, interstitial hyperplasia, and inflammatory infiltrates; the alveoli exhibited a size reduction or fused to form larger alveolar cavities, leading to a decrease in the overall number of alveoli (Figure 6).

In the groups challenged with BoHV-1 HB06, the alveolar structures in all vaccinated groups mainly stayed intact, showing only a minor thickening of the alveolar wall and a small presence of plasmacytic infiltration in the alveolar cavity. Only the alveoli in the 3:3 group fused into larger alveolar cavities after the challenge. In contrast, the unimmunized but challenged group experienced severe lesions, and the alveolar structure was destroyed with interstitial hyperplasia with inflammatory cell exudation (Figure 7). After exposure to *M. bovis* HB0801 and BoHV-1 HB06, the tissue from the control group was sampled and analyzed, and the results displayed a regular alveolar structure, showing clean alveolar cavities and smooth alveolar walls, without any signs of inflammatory exudate. The lung tissue morphology in immunized rabbits was more intact than in unimmunized rabbits. Ultimately, the vaccine provided sufficient protection, particularly in the 2:2 group.

### 3.7. Calculation of Protective Rate of the Attenuated and Marker M. bovis–BoHV-1 Combined Vaccine after Challenge

As described in the Materials and Methods Section, we calculated the protective rate after *M. bovis* HB0801 and BoHV-1 HB06 challenge. As demonstrated in Table 4 and Table 5, the 2:2 immunization dose group provided the best protection when challenged with *M. bovis* HB0801, and its protection rate reached 75.7%. When experimental groups were challenged with BoHV-1 HB06, the 2:2 immunization dose group had the highest protection rate, as high as 87.6%. When facing the challenge of two pathogens, the protection rate of the 2:2 immunization dose group was higher than that of the 1:1 group by 3.9% and 3.7%, respectively, showing the best immunoprotection efficacy.

## 4. Discussion

Bovine respiratory disease (BRD) is considered to be among the most complex mammalian diseases. It involves stress immunosuppression, one or more viral/bacterial infections, and ultimately causes bronchopneumonia. The term bovine respiratory disease can encompass a range of pneumonic diseases, from acute fatal respiratory disease to chronic long-term respiratory disease, in which chronic, necrotizing pneumonia is commonly associated with *M. bovis*, and *M. bovis* is also one of the most common microbial pathogens in cases of lethal BRD [17]. Animal performance and economic efficiency decrease significantly as BRD treatments increase. Furthermore, animals treated with more than one BRD may lack the initial therapeutic effect and develop more severe BRD cases [7]. Although respiratory mucosal epithelial cells are not generally included in the immune system, they play a key role in the first line of defense against infection [10]. Viral pathogens such as BoHV-1 cause ciliary dysfunction, and the temporary immunosuppression induced by it makes animals more susceptible to secondary bacterial infections, leading to BRD. We hypothesized that the coexistence of *M. bovis* with bovine respiratory viruses such as BoHV-1 leads to changes in the respiratory micro-ecological environment. Under these conditions, the virulence of pathogenic microorganisms in vivo also increases.

Vaccination is the most direct and effective way to prevent and control BRD, but challenges remain in vaccine development because BRD is multipathogenic. Therefore, combined and multivalent vaccines must be developed against different pathogens. Several monovalent or multivalent vaccines are currently available to prevent BRD [18,19,20]. Modified live virus (MLV) vaccines induce robust and long-lasting humoral and cellular immunity and only require a smaller dose to provide protection [21,22]. Thus, the MLV vaccine provides better clinical protection against BRD. However, combined live and marker vaccines are not yet available.

In a previous study, we developed an attenuated and marker *M. bovis*–BoHV-1 combined vaccine for the first time. We conducted a comprehensive evaluation of the vaccine using a rabbit model, and the results confirmed its clinical safety and robust protective efficacy [12]. In this study, we further investigated the optimal immunizing dose of the combined vaccine.

Identifying pathogen shedding during the challenging period is pivotal for diagnosing disease [23]. For *M. bovis*, the 1:1 dose of the vaccinated group could effectively inhibit the *M. bovis* HB0801 shedding on the second day after the challenge, and the other doses of the immunized group subsequently failed to detect the shed of *M. bovis* HB0801 by day 3. As for pathogen shedding in the nasal cavity after BoHV-1 challenge, the experimental groups with 1:1 and 3:3 immunization doses displayed deficient shedding levels on the sixth day after challenge, and the vaccinated group with the 2:2 immunization dose had undetectable BoHV-1 shedding on the seventh day after challenge. Our rabbit model results are consistent with previous studies. Following a 2- to 4-day incubation period post-challenge with BoHV-1, viral replication occurs in the mucosal and tonsillar tissues, resulting in clinical manifestations in cattle and subsequent virus shedding within 3 to 10 days post-infection [24]. Subsequently, we assessed antibody titers against BoHV-1 and *M. bovis* and serum-specific IgA and IgG antibody levels. The vaccine-induced antibody response is one of the most important immunological factors against infection. It has also been revealed that gene-deletion vaccines can produce high titers of antibodies after BoHV-1 challenge [25]. sIgA is the most critical antibody for local mucosal immunity. It determines the resistance of the respiratory mucosa to pathogens, whereas systemic humoral immunity depends on IgG and plays an essential role in anti-infective immunity. As expected, when these rabbits vaccinated with different immunizing doses were challenged, high titers of antibodies against both pathogens and high levels of serum-specific antibodies were produced simultaneously, indicating that the attenuated and marker *M. bovis*–BoHV-1 combined vaccine induced a specific humoral immune response along with an effective mucosal immune response. Among all rabbits, the immunized group with a 2:2 immunization dose had the best overall performance, consistent with our speculation that the vaccine’s protective efficacy is positively correlated with the immunization dose. However, the results showed no statistical difference between the 2:2 and the 1:1 immunization dose group.

Moreover, we measured the expression of vaccine-induced cytokines to assess the cellular immune response associated with protection. The homeostatic balance between cytokines plays a key role in the pathogenesis of viral and bacterial infections, including BRD [8,10]. TNF-α is a pro-inflammatory cytokine that can recruit leukocytes to areas of infection [26]. It stands out as one of the key inflammatory mediators essential in initiating and pathogenesis of pulmonary fibrosis [27]. IFN-γ has antiviral effects necessary for innate and adaptive immune responses [28]. Furthermore, the level of IFN-γ secretion was positively correlated with the reduction in clinical signs in infected animals [29]. The production of Th1-type cytokines such as TNF-α and IFN-γ combat intracellular pathogens, and these cytokines also modulate other small-molecule proteins of the immune system through potent antiviral activity and promotion of other immune effector functions [30]. In this study, rabbits immunized with different doses of the vaccine produced high levels of IL-1β, IL-6, TNF-α, and IFN-γ, most notably in TNF-α and IFN-γ. Our attenuated and marker *M. bovis*–BoHV-1 combined vaccine can induce a Th1-biased cellular immune response in animals after immunization. The 2:2 group produced the best immunoprotected response among all immunized groups, but no significant difference existed between all experimental groups.

## 5. Conclusions

In conclusion, the optimal immunization dose of the attenuated and marker *M. bovis*–BoHV-1 combined vaccine was determined in this study; the 2:2 immunization dose group showed the best performance in humoral and cellular immune responses, although there was no significant difference between it and the other immunization dose groups. In addition, from the results of lung histopathological sections and the quantification of immunoprotective rate, the 2:2 vaccinated groups had a nearly standard lung tissue structure and were able to provide the best protective efficacy in facing the challenge with both pathogens. So, we considered that the optimal immunization dose of the vaccine is 2.0 × 10^8^ CFU *M. bovis* HB150 with 2.0 × 10^6^ TCID_50_ BoHV-1 gG-/tk-. One problem was that the 3:3 immunization dose group was not as immunoprotective as the other experimental groups. Our argument stems from the premise that the immunization doses, calibrated according to bovine standards, may have led to an overdose in rabbits. The 3:3 immunization dose group did not produce excellent protective efficacy precisely because of the side effects of overdose immunization. Further bovine experiments are needed to determine the actual clinical application of our study.

## Figures and Tables

**Figure 1 animals-14-00748-f001:**
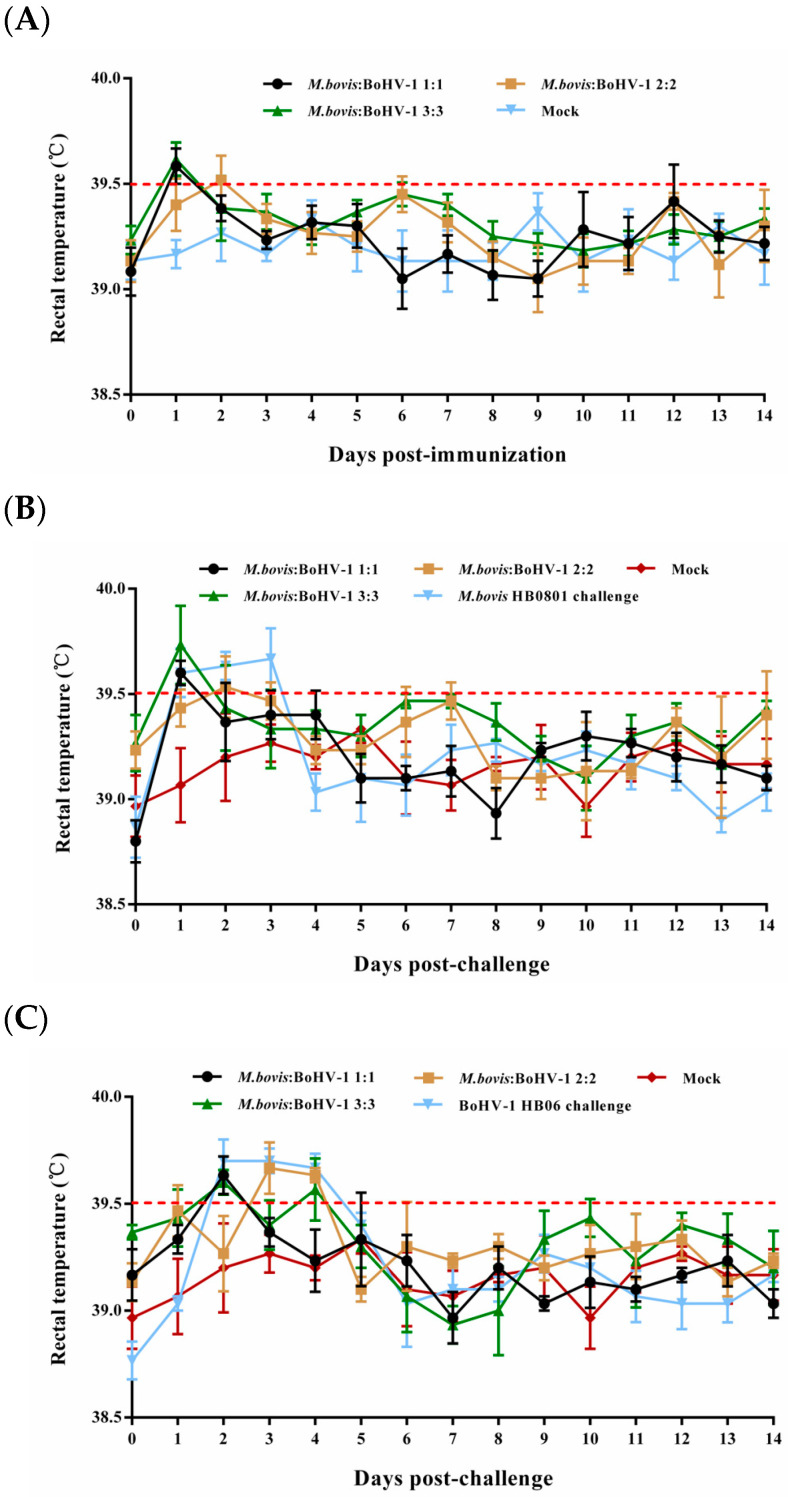
Illustrates the rectal temperature of experimental rabbits before (**A**) and after challenge with *M. bovis* HB0801 (**B**) and BoHV-1 HB06 (**C**). Each data point represents the average rectal temperature of all rabbits in the experimental group on that day, and the red dotted line represents the upper limit of normal rectal temperature of rabbits.

**Figure 2 animals-14-00748-f002:**
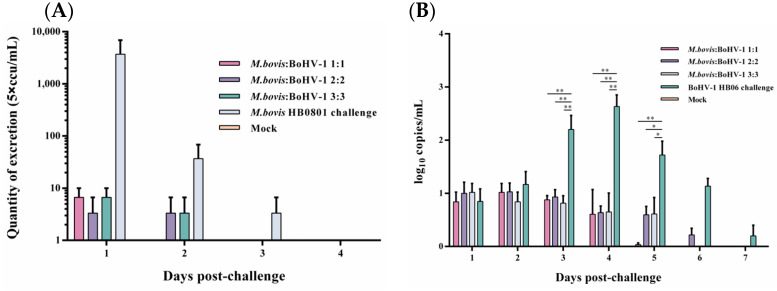
Detection of *M. bovis* HB0801 (**A**) and BoHV-1 HB06 (**B**) shedding in nasal swabs after challenge.

**Figure 3 animals-14-00748-f003:**
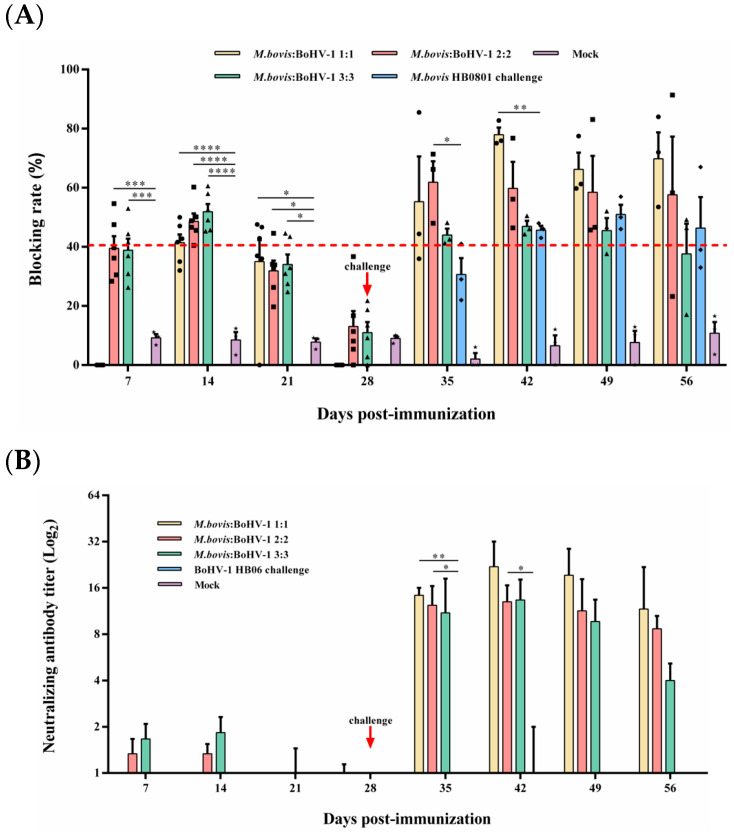
Humoral immune responses induced by *M. bovis*–BoHV-1 combined vaccine. Weekly serum samples were collected for analysis. (**A**) *M. bovis* serum antibody and (**B**) BoHV-1 serum neutralizing antibody.

**Figure 4 animals-14-00748-f004:**
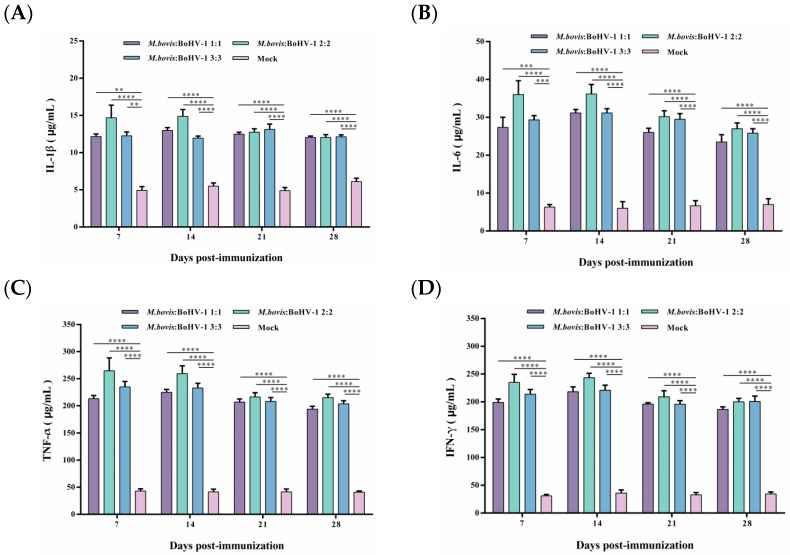
Displays the production of IL-1β (**A**), IL-6 (**B**), TNF-α (**C**), and IFN-γ (**D**) post-immunization. Commercial ELISA kits were utilized to detect these cytokines. The results are presented as means ± SEM.

**Figure 5 animals-14-00748-f005:**
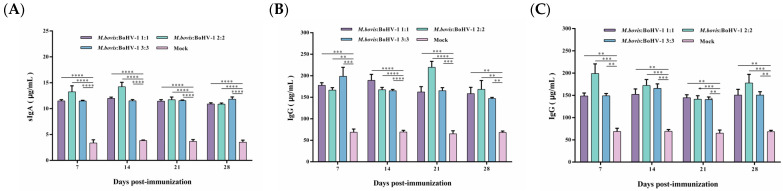
Post-immunization sIgA (**A**) and post-challenge IgG antibody levels after *M. bovis* HB0801 (**B**) and BoHV-1 HB06 (**C**) challenge were monitored.

**Figure 6 animals-14-00748-f006:**
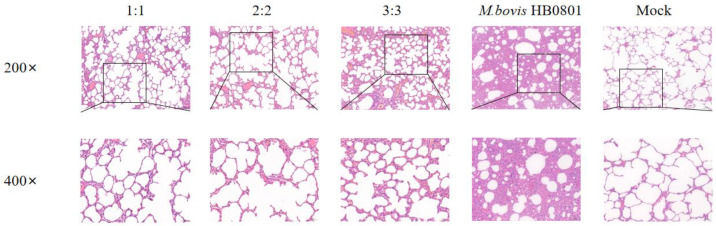
Histopathological images of lung tissues after *M. bovis* HB0801 challenge tainted by H&E. The scale sizes are 100 μm (**top**) and 50 μm (**bottom**), respectively. The figure below is an enlargement of part of the area of the upper figure.

**Figure 7 animals-14-00748-f007:**
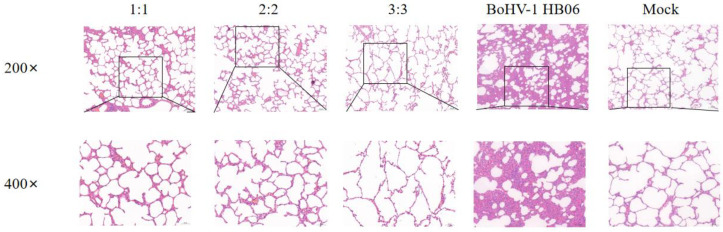
Histopathological images of lung tissues after BoHV-1 HB06 challenge tainted by H&E. The scale sizes are 100 μm (**top**) and 50 μm (**bottom**), respectively. The figure below is an enlargement of part of the area of the upper figure.

**Table 1 animals-14-00748-t001:** Animal immunization and challenge information.

Group	Strain and Dose of Vaccination *	Challenge Strain and Dose **	Number of Rabbits
1	1.0 × 10^8^ CFU *M. bovis* HB150 1.0 × 10^6^ TCID_50_ BoHV-1 gG-/tk-	1.0 × 10^9^ CFU *M. bovis* HB0801	3
2	1.0 × 10^8^ CFU *M. bovis* HB150 1.0 × 10^6^ TCID_50_ BoHV-1 gG-/tk-	4.0 × 10^7^ TCID_50_ BoHV-1 HB06	3
3	2.0 × 10^8^ CFU *M. bovis* HB150 2.0 × 10^6^ TCID_50_ BoHV-1 gG-/tk-	1.0 × 10^9^ CFU *M. bovis* HB0801	3
4	2.0 × 10^8^ CFU *M. bovis* HB150 2.0 × 10^6^ TCID_50_ BoHV-1 gG-/tk-	4.0 × 10^7^ TCID_50_ BoHV-1 HB06	3
5	3.0 × 10^8^ CFU *M. bovis* HB150 3.0 × 10^6^ TCID_50_ BoHV-1 gG-/tk-	1.0 × 10^9^ CFU *M. bovis* HB0801	3
6	3.0 × 10^8^ CFU *M. bovis* HB150 3.0 × 10^6^ TCID_50_ BoHV-1 gG-/tk-	4.0 × 10^7^ TCID_50_ BoHV-1 HB06	3
7	PPLO complete medium	1.0 × 10^9^ CFU *M. bovis* HB0801	3
8	DMEM medium	4.0 × 10^7^ TCID_50_ BoHV-1 HB06	3
9	Blank Control		3

* All vaccines were added dropwise into the nasal cavity using a 2 mL syringe. ** *M. bovis* HB0801 was administered via tracheal injection, while BoHV-1 HB06 was added dropwise through intranasal inoculation.

**Table 2 animals-14-00748-t002:** Detailed primer sequences.

Target Gene	Primer Sequence (5′→3′)	Target Gene Length
*gG*	Forward: CCGACCGCCTCCTACACCAGATGCT	Delated strain: 524 bp Wildtype strain: 1859 bp
	Reverse: GGGTGTAGGCAAGCTCACCGCAACG	
*TK*	Forward: ACGGGCTGGGAAAGACAACAACGG	Delated strain: 235 bp Wildtype strain: 868 bp
	Reverse: GCGGACACGTCCAGCACGAACA	
*gB*	Forward: AGCACCTTTGTGGACCTAA	118 bp
	Reverse: GCTGTATCTCGCTGTAGTCG	
*uvrC*	Forward: TAATTTAGAAGCTTTAAATGAGCGC	238 bp
	Reverse: CATATCTAGGTCAATTAAGGCTTTG	

**Table 3 animals-14-00748-t003:** Shedding of *M. bovis* HB150 and BoHV-1 gG-/tk- after immunization.

Group		Days Post-Immunization (%)								
		1	2	3	4	5	6	7	8	9
1&2 (1:1)	*M. bovis*	6/6 (100)	4/6 (66.7)	4/6 (66.7)	1/6 (16.7)	0/6 (0)	0/6 (0)	0/6 (0)	0/6 (0)	0/6 (0)
	BoHV-1	6/6 (100)	6/6 (100)	5/6 (83.3)	6/6 (100)	6/6 (100)	3/6 (50)	2/6 (33.3)	1/6 (16.7)	0/6 (0)
3&4 (2:2)	*M. bovis*	6/6 (100)	6/6 (100)	4/6 (66.7)	4/6 (66.7)	2/6 (33.3)	0/6 (0)	0/6 (0)	0/6 (0)	0/6 (0)
	BoHV-1	6/6 (100)	6/6 (100)	6/6 (100)	5/6 (83.3)	4/6 (66.7)	2/6 (33.3)	2/6 (33.3)	0/6 (0)	0/6 (0)
5&6 (3:3)	*M. bovis*	6/6 (100)	5/6 (83.3)	6/6 (100)	4/6 (66.7)	3/6 (50)	2/6 (33.3)	0/6 (0)	0/6 (0)	0/6 (0)
	BoHV-1	6/6 (100)	5/6 (83.3)	6/6 (100)	6/6 (100)	4/6 (66.7)	3/6 (50)	1/6 (16.7)	2/6 (33.3)	0/6 (0)
9 (mock)		0/3	0/3	0/3	0/3	0/3	0/3	0/3	0/3	0/3

The number above each cell represents the number of rabbits per group that was able to detect the shedding of bacteria or viruses/the total number of rabbits per group; the percentage in brackets at the bottom represents the percentage of the total number of rabbits within each group that could detect bacterial/viral shedding.

**Table 4 animals-14-00748-t004:** Protective rate after *M. bovis* HB0801 challenge.

Group	Average Score		Protective Rate (%)
	Clinical Score	Lung Lesion Score	
*M. bovis*–BoHV-1 1:1	0.16	5.09	71.8%
*M. bovis*–BoHV-1 2:2	0.12	4.67	75.7%
*M. bovis*–BoHV-1 3:3	0.14	5.33	69.9%
*M. bovis* HB0801 challenge	0.84	12.67	\
Mock	0	2.00	\

**Table 5 animals-14-00748-t005:** Protective rate after BoHV-1 HB06 challenge.

Group	Average Score		Protective Rate (%)
	**Clinical Score**	**Lung Lesion Score**	
*M. bovis*–BoHV-1 1:1	0.27	3.75	83.9%
*M. bovis*–BoHV-1 2:2	0.22	3.33	87.6%
*M. bovis*–BoHV-1 3:3	0.31	4.33	79%
BoHV-1 HB06 challenge	1.3	13.25	\
Mock	0	2.00	\

## Data Availability

Data are contained within the article.

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
