# Peer review of "Investigation of the Optimal Immunization Dose and Protective Efficacy of an Attenuated and Marker M. bovis–Bovine Herpesvirus Type 1 Combined Vaccine in Rabbits"

_animals, 2024, doi:10.3390/ani14050748_

Round 1
Reviewer 1 Report
Comments and Suggestions for Authors
I read the manuscript entitled "Investigation of the optimal immunization dose and protective efficacy of an attenuated and marker M. bovis-BoHV-1 combined vaccine in rabbits" with great interest. I found it a logical continuation of the earlier works of this team of authors.
However, while reading, I had a number of comments about this manuscript.
The main remark concerns the "Introduction" section. In my opinion, in this section the authors should describe existing vaccines in more detail and justify the potential benefits of the vaccine they developed. It would also be desirable to justify the choice of an animal model with at least a few sentences. It is unclear from the text of the manuscript why a potentially lethal infection is being studied specifically in rabbits, in which the disease did not cause death.
Less significant comments
1) line 146 "Genomes of nasal swabs...."
It seems to me that this is not a completely correct formulation, and we are not talking about genomes, but rather about preparations of total nucleic acids
2) The group designations 1:1, 1:2 and 3:3 are not intuitive. In addition, the question arises: why were such doses and such ratios of vaccine components chosen?
3) The text does not clearly indicate how exactly the authors carry out a quantitative assessment of such indicators as “clinical score” and “Lung lesion score”.
4) There are no spaces in the entire text of the manuscript before the sign [
Author Response
Dear reviewer, since only one file can be submitted on the website, I have sent the revised manuscript to the editorial department by email. Please check.

Reviewer 2 Report
Comments and Suggestions for Authors
Overall this appears to be a relatively sound study. Rabbits were vaccinated with a range of vaccine doses and challenged 28 days after vaccination. The authors observed a range of parameters and the conclusions appear to be a relatively sound.
However, the language used in this paper leaves a little to be desired.
For example in line 23 in the abstract it was noted that rabbits that received immunisation showed increased humeral and cellular immunological reactions. Animals do not show reactions. Samples are collected from the animals and observations are made. Conclusions are then drawn from this.
The authors could have written that assays performed on the samples collected from immunised animals indicated that there were increased humeral and cellular immunological reactions.
Comments on the Quality of English LanguageLine 40
The word “cattle” in this case is inappropriate. These may be cows or is there are significantly different age groups bovids.
Line 63 is an example of inappropriate terminology. The vaccines do not show remarkable efficiency. This paragraph needs to be rewritten to indicate that the vaccines were relatively efficient. However, the vaccine is not capable of showing efficiency.
Line 81 refers to a previous study. There should be a reference to this study. There is no indication that the cell cultures that were used were free from adventitious agents such as BVD and Mycoplasma. There is no indication that the foetal bovine serum is certified free of BVD. This may have been discussed in the previous study. If it was discussed in the previous study than a brief mention of it would be helpful at this stage. If this was not discussed in the previous study the methods used to determine freedom from adventitious agents should be added to this paragraph.
Line 146
The authors refer to using RT-PCR. They then go on to describe PCR not RT-PCR. RT-PCR is the abbreviation for reverse transcriptase PCR that is carried out to determine the presence of specific RNA. It would appear that only DNA is being detected and there is no reverse transcriptase step described in the assays.
I assume that the indicator system was electrophoresis of the samples in an agarose gel. This has been omitted.
The assay that is described is only qualitative not quantitative and perhaps in the discussion the authors could mention that the addition of a quantitative assay such as TaqMan could have allowed the results to be quantitative and provide additional information.
Line 164
PCR not RT-PCR
Lines 163 – 164
If the method was described previously a reference should be added. The description of the method in this paragraph is sufficient and there is possibly no need to mention that this has been previously described.
Line 225
It was indicated that animals exhibited continuous shedding. This should be rewritten to show that samples collected from these animals over this period contained M. bovis as determined by PCR
Similar comments apply to lines 228 and 230.
Similarly in line 241 it was indicated that the immunised animals displayed a decrease in M. bovis.
And in 247 that the immunised group exhibited lower shedding.
This sort of terminology is seen throughout the text and a major rewrite is required to indicate that:
· Samples were collected
· assays were carried out
· observations were made
· conclusions were drawn
Likewise in line 345 it was noted that the control group exhibited such normal alveoli structure. A normal alveoli structure was observed in histological sections and these were taken from animals in the control group. The terminology used is quite unacceptable as the animals cannot exhibit the absence or presence of a lesion. These lesions are observed and recorded.
In line 362 and 363 the authors indicated that some immunisation groups showed the best immune protection efficiency. The groups do not indicate this. However, there is a correlation with immunisation efficiency and specific treatments. These paragraphs need to be rewritten.
Lines 379 to 382 describe an important attribute of a discussion. This is an excellent guideline but there is no need for it to be included in the paper.
Line 384 there is no need for the word attack
Line 410
The word “good” refers to behaviour not to a value. It might be effectively protective or there could be sufficient protection but the word good is not appropriate.
This paragraph refers to a previous study. However, there is no reference to the previous study. This should be added.
Author Response

(The authors gave the same response as above.)

Reviewer 3 Report
Comments and Suggestions for Authors
The authors present an interesting study where they have evaluated the capacity of a combined bovine herpesvirus 1 (BoHV-1) and Mycoplasma bovis vaccine formulation in the context of bovine respiratory disease (BRD). The modified live vaccine formulation was evaluated in a rabbit model. Strong cell-mediated and humoral responses were detected in the immunised rabbits compared to the controls.
The presented data supports the conclusions drawn.
Lines 40 to 43 - I would suggest citing all of these values in USD to enable direct comparison of the costs. All monetary values should also have a "$".
Line 45 suggest revision "also, the susceptibility of cattle to this disease can be influenced"
Line 48 suggest revision "also significantly associated with the risk of severe secondary bacterial infections."
Causality can be very difficult to attribute in diseases such as BRD.
A suitable citation would be useful for this statement.
Line 51 suggest revsion "are the pathogens most frequently associated with BRD in China."
Please provide a suitable citation for this statement.
Line 55 Please clarify what type of prevalence is referred to here. For example is it seroprevalence?
Line 107 Suggest adding another column with the number of animals in each treatment group.
Line 129 suggest revision "All vaccines were added dropwise to the"
If this accurately describes the process used.
Line 131 Was the BoHV-1 challenge delivered dropwise or aerosolised?
Lines 204-212 These results are mostly descriptive. Were any of the differences observed between the groups statistically significant?
Line 219 The figure legend should be revised to ensure that the figure can be interpreted in the absence of the main text.
This should include:
What do the data points represent?
What do the error bars represent?
What is the red dotted line for?
I would also suggest the authors consider splitting the graphs into four panels so that each treatment group is plotted with the control group. In its current format, many of the data points and connecting lines are superimposed on each other.
I realise this would dramatically increase the number of figures in the manuscript, however, if the figure does not clearly illustrate the data then it is of limited value. The authors may need to consider if some of the data could be provided as supplemental files.
Line 233 The table title should be expanded to clearly describe the data shown.
Line 254 - See comments for Figure 1, for the need for the content to be better described.
Same comments for the remaining figures.
Line 336 No scale bars are shown in the panels.
Line 351 No scale bars are shown in the panels.
Lines 379-382 - suggest deletion.
Author Response

(The authors gave the same response as above.)
